# CW-CNN & CW-AN: Convolutional Networks and Attention Networks for CW-Complexes

## Abstract

We present a novel framework for learning on CW-complex structured data points. Recent advances have discussed CW-complexes as ideal learning representations for problems in cheminformatics. However, there is a lack of available machine learning methods suitable for learning on CW-complexes. In this paper we develop notions of convolution and attention that are well defined for CW-complexes. These notions enable us to create a Hodge informed neural network that can receive a CW-complex as input. We illustrate and interpret this framework in the context of supervised prediction.

## 1 Introduction

### 1.1 Complexes

Succinctly, a cell complex is an object in a category obtained by successively gluing together cells using pushouts. More formally, Whitehead (1949) defined them in the following way.

**Definition 1.1.** A cell complex $K$, or alternatively a *complex*, is a Hausdorff space which is the union of disjoint open cells $e, e^n, e_i^n$ subject to the condition that the closure $\bar{e}^n$ of each $n$-cell, $e^n \in K$ is the image of a fixed $n$-simplex in a map $f : \sigma^n \to \bar{e}^n$ such that

(1) $f|\sigma^n - \partial\sigma^n$ is a homeomorphism onto $e^n$

(2) $\partial e^n \subset K^{n-1}$, where $\partial e^n = f\partial\sigma^n = \bar{e}^n - e^n$ and $K^{n-1}$ is the $(n-1)$-section of $K$ consisting of all the cells whose dimensionalities do not exceed $n - 1$.

A CW-complex is a cell complex that has the weak topology and is closure finite. A complex $K$ is said to be closure finite if and only if $K(e)$ is a finite subcomplex, for every cell $e \in K$. We say $K$ has the weak-topology if and only if a subset $X \subset K$ is closed provided $X \cap \bar{e}$ is closed for each cell $e \in K$. To construct a CW-complex, we inductively glue cells together. More formally, Hatcher (2002) describes how we construct a finite CW-complex $X$ as follows. Initially, we start with a collection of zero cells $X^0 = \{e_i^0\}_{i=0}^N$. $X^0$ is called the 0-skeleton. Then, for all $j \in \{1, \ldots, n\}$ we take a collection of $j$-cells $\{e_i^j\}_{i=0}^N$ and glue their boundaries to points in the $j - 1$ skeleton using continuous attaching maps $\phi_i^j : \partial e_i^j \to X^{j-1}$. Each $j$-cell is a topological space. Essentially, a CW-complex is constructed by taking a union of a sequence of topological spaces $\varnothing = X^{-1} \subset X^0 \subset X^1 \subset \cdots$ such that each $X^j$ is obtained by attaching $j$-cells to $X^{j-1}$. In the language of category theory, we often think of the topology on finite CW-complex $X$ as the direct limit of the diagram $X^{-1} \hookrightarrow X^0 \hookrightarrow X^1 \hookrightarrow \cdots \hookrightarrow X^k$ for some $k \in \mathbb{N}$. CW-complexes generalize the notion of a graph. A 1-dimensional CW-complex is a regular graph without loops. Moreover, every topological space is weakly homotopy equivalent to a CW-complex.

### 1.2 Learning on CW-complexes

Consider the following learning problem. Suppose we are presented with a dataset $\mathcal{D} = \{(x_i, y_i)\}_{i=1}^n$ where $x_i$ is a CW-complex, $y_i \in \mathbb{R}^d$, and $n, d \in \mathbb{N}$. Then, the task of learning a function $\mathcal{F}$ such that $y_i = \mathcal{F}(x_i) + \epsilon$,

where $\epsilon$ is some error, naturally arises. We tackle this problem by developing a convolutional layer and attention mechanism for a CW-complex $x_i$. Essentially, we extend the work of Kipf & Welling (2017) to define a notion of convolution for CW-complexes. Additionally, we extend the work of Veličković et al. (2018) to develop a notion of attention for CW-complexes.

## 2 Related Work

### 2.1 Graph Neural Networks

Kipf & Welling (2017) develop a semi-supervised learning framework for graphs. The authors consider a graph-based semi-supervised learning approach, where label information is smoothed over the graph by applying a form of explicit graph-based regularization. The authors introduce a well behaved layer-wise propagation rule, and demonstrate semi-supervised classification of graphs.

### 2.2 Graph Attention Networks

Veličković et al. (2018) develop a notion of attention for Graphs. Let $\mathcal{G} = (\mathcal{V}, \mathcal{E})$ contains nodes $\mathcal{V} = \{1, \ldots, n\}$ and edges $\mathcal{E} \subseteq \mathcal{V} \times \mathcal{V}$, where $(j, i) \in \mathcal{E}$ denotes an edge from a node $j$ to a node $i$. We assume that every node $i \in \mathcal{V}$ has an initial representation $h_i^0 \in \mathbb{R}^{d_0}$. A Graph Neural Network takes in a set of node representations $\{h_i \in \mathbb{R}^d \mid i \in \mathcal{V}\}$ and the set of edges $\mathcal{E}$ as input. The layer outputs a new set of node representations, where the same parametric function is applied to every node given its neighbors $\mathcal{N}_i = \{j \in \mathcal{V} \mid (j, i) \in \mathcal{E}\}$. A Graph Attention Network computes a learned weighted average of the representations of $\mathcal{N}_i$. A scoring function $e : \mathbb{R}^d \times \mathbb{R}^d \to \mathbb{R}$ computes a score for every edge $(j, i)$, which indicates the importance of the features of the neighbor $j$ to node $i$. Then the Graph Attention Network computes a new node representation using a chosen nonlinearity (Veličković et al., 2018; Brody et al., 2022).

### 2.3 Gaussian Processes on Cellular Complexes

Alain et al. (2023) define the first Gaussian process on cell complexes. In doing so, the authors define the Hodge Laplacian and introduce important notation. We reproduce these definitions in the appendix A.1. We reference these concepts throughout the paper.

### 2.4 A Comparison of CW-Complex Networks

Recent advances have proposed neural networks for CW-complexes. In particular, Giusti et al. (2023) developed cell attention networks. Additionally, both Hajij et al. (2021) and Bodnar et al. (2021) develop message passing schemes for CW-complexes. In this section we provide a theoretical and intuitive comparison of CW-complex networks.

#### 2.4.1 Theoretical Comparisons

In the following subsection we break down theoretical factors distinguishing each model. We focus on time complexity, number of network parameters, invariances, and hodge-awareness. Through this theoretical comparison we can get a sense of the scalability and properties of the various networks.

**Time Complexity**  As described by Blakely et al. (2021), in the context of graph neural networks the traditional message passing algorithm, for dense graphs, has forward pass and backward pass time complexity of $O(LN^2F + LNF^2)$ where $L$ is the number of layers, $N \times N$ is the size of the graph adjacency matrix, and $F \times F$ is the size of the weight matrix. Other authors leverage the $k$-WL hierarchy to state that 3-WL equivalent graph neural networks have time complexity $O(n^3)$ with $O(n^2)$ space complexity (Balcilar et al., 2021). We follow the first convention. In particular, Let $M$ be the number of neural networks one wishes to train. Let $L$ be the number of layers per network. Let $O(K^2)$ be the size of representing a particular dense/well-connected $k$-chain. Let the size of the $n$-th weight matrix be $O(F^2)$. Let $S$ be the time-complexity of applying a lifting map. Instead of reporting both backward and forward pass time complexities, we simply report the worst case time-complexity as one has to run both to train a network.

**Network Parameters** The parameters in a network are undoubtedly crucial. However the trainable parameters are not necessarily limited to only weight matrices. For a weight matrix of size $F \times F$, we say there are $O(F^2)$ many entries which are trainable. We say a trainable function, for example the $\phi_j^{(k)}$ in Hajij's scheme which is in practice an MLP, has $O(P)$ many parameters Hajij et al. (2021). For reference, note that usually $P >> F^2$ where $F^2$ is as defined above.

**Mechanism** There are numerous mechanisms utilized. We report two values in this column. One for the kind of network, either attention or convolutional, and the other for the core method by which one propagates information or passes messages through the network.

**Invariances/Equivariances** We follow the usual definition of both and report any kinds of invariance or equivariance present in the networks. One can review Maron et al. (2018) for a detailed account.

**Hodge Informed** The traditional definition of Hodge awareness strictly requires several conditions and was originally defined for simplical complex networks (Yang & Isufi, 2023). In our definition for CW-complex networks, we relax exactly one condition namely that learning in the gradient and curl spaces do not necessarily have to be independent. A CW-complex network is defined to be Hodge-aware if:

(1) The filter/matrix $\mathbf{H}_k$ is a Hodge-invariant learning operator. Specifically, three Hodge subspaces are invariant under $\mathbf{H}_k$.

(2) The lower filter $\mathbf{H}_{k,d}$ and upper filter $\mathbf{H}_{k,u}$ are, respectively, gradient and curl-invariant learning operators. This notation is analogous to lower and upper triangles of a matrix $\mathbf{H}_k$.

(3) The learning in the gradient and curl spaces are expressive.

Under this definition our proposed networks are Hodge aware. However, message passing networks which rely on an MLP to aggregate and update are not Hodge aware (Anonymous, 2024). Bodnar et al. (2021), Hajij et al. (2021), and Giusti et al. (2023) all rely on an MLP for their aggregation or update steps in practice. Consequently, none of their networks are Hodge aware.

Table 1: Theoretical comparisons

| Model | Time Complexity | Parameters | Mechanism | Invariances | Hodge Informed |
|---|---|---|---|---|---|
| CWN Bodnar et al. (2021) | $O(LK^2P)$ | $O(LP)$ | convolution, lifting maps+MLP | permutation | ✗ |
| CXN (Hajij et al., 2021) | $O(MLK^2P^2)$ | $O(MLP^2)$ | convolution, MLP | permutation | ✗ |
| CAN Giusti et al. (2023) | $O(LK^2P^2S)$ | $O(LP^2)$ | attention, lifting maps+MLP | permutation | ✗ |
| **CW-AT** | $O(LK^2F^2)$ | $O(LF^2)$ | attention, boundary operators+summation | hodge | ✓ |
| **CW-CNN** | $O(LK^2F^2)$ | $O(LF^2)$ | convolution, boundary operators | hodge | ✓ |

### 2.4.2 Intuitive Comparisons

**CW Networks (CWN)** Bodnar et al. (2021) extend a message-passing algorithm to cell complexes. Bodnar et al. (2021) essentially define lifting transformations, $f : G \rightarrow X$, augmenting a graph with higher-dimensional constructs. This results in a multi-dimensional and hierarchical message passing procedure over the input graph (Bodnar et al., 2021). The authors specify this procedure over the space of chemical graphs in section 4, defining message passing from atoms to bonds, and bonds to rings.

**CWN Message Passing** Bodnar et al. (2021) allow their CWN to receive two kinds of messages

$$m_{\mathcal{B}}^{t+1}(\sigma) = \text{AGG}_{\tau \in \mathcal{B}(\sigma)} \left( M_{\mathcal{B}} \left( h_\sigma^t, h_\tau^t \right) \right) \quad m_\uparrow^{t+1}(\sigma) = \text{AGG}_{\tau \in \mathcal{N}_\uparrow(\sigma), \delta \in C(\sigma,\tau)} \left( M_\uparrow(h_\sigma^t, h_t^t, h_\delta^t) \right) \tag{1}$$

The first kind of specifies messages from atoms to bonds and from bonds to rings. The second kind specifies messages between atoms that are connected by a bond and messages between bonds part of the same ring.

The update operation takes into account these two types of messages and updates features of the cells by applying the rule:

$$h_\sigma^{t+1} = U\left(h_\sigma^t, m_\mathcal{B}^t(\sigma), m_\uparrow^{t+1}(\sigma)\right) \tag{2}$$

One can obtain a global embedding of a cell complex $X$, using a model with $L$ layers via a readout function which takes as input the separate multi-sets of features

$$h_X = \text{READOUT}\left(\{\{h_\sigma^L\}\}_{\dim(\sigma)=0}, \{\{h_\sigma^L\}\}_{\dim(\sigma)=1}, \{\{h_\sigma^L\}\}_{\dim(\sigma)=2}\right) \tag{3}$$

As described by in Appendix E.3 by Bodnar et al. (2021) throughout all experiments, the cellular message passing layers update the representation of $p$-cell $\sigma$ as follows:

$$h_\sigma^{t+1} = \text{MLP}_{U,p}^t\left(\text{MLP}_{\mathcal{B},p}^t((1+\epsilon_\mathcal{B})h_\sigma^t + \sum_{\tau\in\mathcal{B}(\sigma)} h_\tau^t)\|\text{MLP}_{\uparrow,p}^t((1+\epsilon_\uparrow)h_\sigma^t + \sum_{\tau,\delta} \text{MLP}_{M,p}^t(h_\tau^t\|h_\delta^t))\right) \tag{4}$$

**CWN Comparison**   In this manuscript we propose a network that receives as input a CW-complex of dimension $n \geq 1$, which need not be a graph. Additionally we do not rely on lifting transformations or any MLP's for aggregation. This results in lower time complexity, and fewer parameters. Moreover due to use of boundary operators, we develop a more computationally efficient way to propagate information through the network. Our method has the additional theoretical benefits of being Hodge aware.

**Cell Complexes Neural Networks (CXN)**   Hajij et al. (2021) propose an inter-cellular message-passing scheme on cell complexes. Under the proposed scheme, the propagation algorithm then performs a sequence of message passing executed between cells in $X$. For all $k \in (0, L]$, and $j \in [0, n)$ where $n, L \in \mathbb{N}$, the forward propagation scheme is defined as

$$h_{c^j}^{(k)} := \alpha_j^{(k)}\left(h_{c^j}^{(k-1)}, E_{a^j\in\mathcal{N}_{adj}(c^j)}\left(\phi_j^{(k)}(h_{c^j}^{(k-1)}, h_{a^j}^{(k-1)}, F_{e^n\in\mathcal{CO}[a^j,c^j]}(h_{e^{j+1}}^{(0)}))\right)\right) \tag{5}$$

where $h_{e^m}^{(k)}, h_{a^m}^{(k)}, h_{c^m}^{(k)} \in \mathbb{R}^{\ell_m^k}$, $E$, and $F$ are permutation invariant differentiable functions and $\alpha_j^{(k)}$, $\phi_j^{(k)}$ are trainable differentiable functions, in essence MLP's.

**CXN Message Passing**   Equation (5) described by Hajij et al. (2021) defines the CXN message passing scheme. One chooses a desired network depth $L$ and $n$ corresponds to the dimension of the cell complex $X$. In practice and in the experiments, the $\alpha_j^{(k)}$, $\phi_j^{(k)}$ are MLP's or convolutional layers (Hajij et al., 2021). However, the above scheme does not necessarily need to be parameterized to MLP's or convolutional layers. The scheme presented provides an effective generalization of the notions of message passing schemes in graphs. The mathematical definition allows for general selection of the $\alpha_j^{(k)}$ and $\phi_j^{(k)}$ provided they are differentiable and trainable functions.

**CXN Comparison**   As stated by Hajij et al. (2021), one may wish to train a CCXN for every $k$-cells adjacency matrix individually. Consequently, one has to train $M = n - 1$ many networks. In contrast to CXN, our models do not rely on MLP's for aggregation. We do not need to train $M = n - 1$ many networks. In fact, we propose training only one network. This results in significantly lower time complexity, and significantly fewer parameters. Moreover due to use of boundary operators, we develop a more computationally efficient way to propagate information through the network. This results in overall much better scaling potential. Our method has the additional theoretical benefits of being Hodge aware.

**Cell Attention Network (CAN)**   Giusti et al. (2023) introduce a neural architecture operating on data defined over the vertices of a graph. The approach described leverages a lifting algorithm that learns edge features from node features, then applies a cellular attention mechanism, and finally applies pooling. In particular, Giusti et al. (2023) define a cellular lifting map as a skeleton-preserving function $s : G \to C_G$ incorporating graph $G$ into regular cell complex $C_G$. Using the cellular lifting map the authors define attentional lift, giving way to their attention mechanism. The procedure computes $F^0$ many attention heads

such that when given input graph $G = (\mathcal{V}, \mathcal{E})$, for vertices $i, j \in \mathcal{V}$ connected by edge $e \in \mathcal{E}$, edge features $x_e \in \mathbb{R}^{F^0}$ are computed by concatenating attention scores. Mathematically written as

$$x_e = g(x_i, x_j) = \|_{k=1}^{F^0} a_n^k(x_i, x_j), \ \forall \ e \in \mathcal{E} \tag{6}$$

Giusti et al. (2023) utilize equation (2) above to define their layer propagation scheme by combining the attentional lift with message passing and performing the aggregation using learnable functions, $\psi_u^l, \psi_d^l$, and an MLP (Giusti et al., 2023).

**CAN Message Passing**   Giusti et al. (2023) describe a Cellular Attentional Message-passing scheme. For each layer $l \in \{1, \ldots, L\}$ the model performs an edge pooling operation after each round of message passing thereby producing a sequence of cell complexes $\{C^l\}_l$ such that $\mathcal{C}^{l+1} \subseteq C^l$. The message passing scheme happens at the edges level in each layer $l$, thereby exploiting upper and lower edge adjacencies $\mathcal{N}_d^l(e)$ and $\mathcal{N}_u^l(e)$ respectively. At each layer $a_u^l$ is a learnable upper attention function and $a_d^l$ is a lower attentional function. Each kind of function evaluates the reciprocal importance of two edges part of the same polygon. The message passing equation then follows as

$$\tilde{h}_e^l = \phi^l \left( h_e^\ell, \bigoplus_{k \in \mathcal{N}_d^l(e)} a_d^l(h_e^l, h_k^l) \psi_d^l(h_k^l), \bigoplus_{k \in \mathcal{N}_u^l(e)} a_u^l(h_e^l, h_k^l) \psi_u^l(h_k^l) \right) \tag{7}$$

Where $\bigoplus$ is any aggregation operator (sum, mean, max, etc.) and $\phi^l$ is a possibly learnable function, and $\psi_u^l$, $\psi_d^l$ are learnable functions sharing the attention weights with $a_u^l$ and $a_d^l$ respectively. Then, after each message passing step, the edge pooling operation occurs. Given that $\tilde{h}_e^l$ is the hidden feature vector associated to edge $e$, the attention pooling operation consists of computing a self-attention score $\gamma_e^l$ for each edge of the complex via a pooling learnable attention function $a_p^l$. In essence $\gamma_e^l = a_p^l(\tilde{h}_e^l)$. Let $k$ be the pooling ratio, in essence the fraction of edges retained after being put through the edge pooling layer. The top-$k$ highest self-attention scoring edges are kept after the pooling stage. In essence $\mathcal{E}^{l+1} = \{e \mid e \in \mathcal{E}^l \wedge \gamma_e^l \in \text{top}-k(\{\gamma_e^l\}_{e \in \mathcal{E}^l}, \lceil k|\mathcal{E}^l|\rceil) \subseteq \mathcal{E}^l$ The feature vectors that are kept after pooling are scaled as $h_e^{l+1} = \gamma_e^l \tilde{h}_e^l$. A consequence of this pooling operation is one needs to adjust the structure of the cell complex $C^l$ in order to obtain a consistent updated complex $C^{l+1}$. The hierarchical version of this can be summarized as a by-layer readout operation on the $\{h_e^{l+1}\}_{e \in \mathcal{E}^{l+1}}$ to obtain an aggregate embedding of the whole complex $C^{l+1}$ as

$$h_{C^{l+1}} = \bigoplus_{e \in \mathcal{E}^{l+1}} h_e^{l+1} \tag{8}$$

Then after the last hidden layer, a global readout operation is performed aggregating the previously computed complexes:

$$h_C = \bigoplus_l h_{C^l} \tag{9}$$

Finally, this aggregation $h_C$, is fed to a multi-layer perceptron (MLP) for the learning task.

**CAN Comparison**   Our models do not rely on MLP's/learnable functions for aggregation. We do not incorporate an input graph $G$ into a regular cell complex $C_G$ or attentional lift. We do not leverage edge-pooling or modify the structure of the CW complex as information propagates through the network. Moreover due to use of boundary operators, we develop a more computationally efficient way to propagate information through the network. Additionally by propagating over cells via the boundary operator, our method can account for the topology of the individual cell and its open neighborhoods. This results in significantly lower time complexity, and significantly fewer parameters. Consequently, overall our models have much better scaling potential. Our method has the additional theoretical benefits of being Hodge aware. Finally, our method is leveraged to develop multi-head attention.

## 3 Hodge-Laplacian informed CW-Complex Networks

### 3.1 Convolutional CW-complex Layer

**Definition 3.1.** Let $X$ be a finite $n$-dimensional CW-complex. Then for $k \in [0, n]$ let $A_k \in \mathbb{R}^{N_k \times N_k}$ be a matrix of cell feature vectors and let $\Delta_k$ be the Hodge Laplacian. Then we can define a **Convolutional CW-complex Network (CW-CNN)** $f(X)$ as being composed by stacking hidden layers $H^{(k)}$ according to the following layer-wise propagation rule:

$$H^{(k+1)} = \sigma \left( B_{k+1}^\top \left( \Delta_k A_k H^{(k)} \right) B_{k+1} \right) \tag{1}$$

Initially, we set $H^{(0)} = X^0 \in \mathbb{R}^{N_0 \times N_0}$, which is the matrix representation of the zero-skeleton of CW-complex $X$. Recall the definitions of the boundary operator (definition A.2), coboundary operator (definition A.4), and Hodge-Laplacian (definition A.6). By definition A.6, $\Delta_k = B_k^\top W_{k-1}^{-1} B_k W_k + W_k^{-1} B_{k+1} W_{k+1} B_{k+1}^\top$. Additionally, by definitions A.2 and A.4, $B_k \in \mathbb{Z}^{N_{k-1} \times N_k}$, $B_{k+1}^\top \in \mathbb{Z}^{N_{k+1} \times N_k}$, and the weight matrix $W_k = \text{diag}(w_1^k, \ldots, w_{N_k}^k) \in \mathbb{R}^{N_k \times N_k}$.
Checking the dimensions we can see that $\Delta_k \in \mathbb{R}^{N_k \times N_k}$:

$$\dim(B_k^\top W_{k-1}^{-1} B_k W_k) = (N_k \times N_{k-1})(N_{k-1} \times N_{k-1})(N_{k-1} \times N_k)(N_k \times N_k) = N_k \times N_k \tag{2}$$

$$\dim(W_k^{-1} B_{k+1} W_{k+1} B_{k+1}^\top) = (N_k \times N_k)(N_k \times N_{k+1})(N_{k+1} \times N_{k+1})(N_{k+1} \times N_k) = N_k \times N_k \tag{3}$$

Therefore $\Delta_k \in \mathbb{R}^{N_k \times N_k} \implies \dim(\Delta_k A_k) = N_k \times N_k$. Additionally, we know from above $\dim(B_{k+1}) = N_k \times N_{k+1}$ and $\dim(B_{k+1}^\top) = N_{k+1} \times N_k$. Therefore, by induction, we can show $\dim(H^{(k)}) = N_k \times N_k$ (Lemma 3.2). Formally, we call $B_k$ the order $k$ incidence matrix, and let $\sigma$ be any nonlinearity. Thus, using the layer-wise propagation rule from equation (1), we can define a neural network $f(X)$ by stacking the hidden layers $H^{(k)}$. We call such a network a Convolutional CW-complex Network, or CW-CNN for short.

**Lemma 3.2.** *The dimension of hidden layer $k$ in a CW-CNN is $\dim(H^{(k)}) = N_k \times N_k$.*

*Proof.* We want to show that $\dim(H^{(k)}) = N_k \times N_k$. For the base case $(k = 0)$ we define $H^{(0)} = X^0 \in \mathbb{R}^{N_0 \times N_0}$. Let the inductive hypothesis $P(j)$ be that $\forall j \in \{0, 1, \ldots, k-1\}$ $\dim(H^{(j)}) = N_j \times N_j$. Then, we can show $P(j) \implies P(j+1)$ using equation (1). We know $\dim(H^{(j+1)}) = \dim(\sigma(B_{j+1}^\top(\Delta_j A_j H^{(j)})B_{j+1})) = (N_{j+1} \times N_j)(N_j \times N_j)(N_j \times N_j)(N_j \times N_j)(N_j \times N_{j+1}) = N_{j+1} \times N_{j+1}$. Thus we see $\dim(H^{(k)}) = N_k \times N_k$. $\square$

The weight matrices $W_k$ can be randomly initialized by choosing $w_i^k$ randomly or by adopting a similar strategy to He Initialization (He et al., 2015). In order to train this network, one would update the weight matrices $W_k$ with gradient descent and define a loss function $\mathcal{L}$. One can replace $W_k^{-1}$ by $W_k^\dagger$, the Moore-Penrose pseudoinverse of $W_k$, in the expression for $\Delta_k$. Then, let $\mathcal{D} = \{(\mathbf{x}_i, \mathbf{y}_i)\}_{i=1}^M$ be a dataset where $\mathbf{x}_i$ is a CW-complex and $\mathbf{y}_i \in \mathbb{R}^d$ where $d \in \mathbb{N}$. Our learning paradigm then becomes $\mathbf{y}_i = f(\mathbf{x}_i) + \varepsilon$, where $f$ is a CW-CNN. If we let $\mathbf{X} = [\mathbf{x}_1, \ldots, \mathbf{x}_M]$ and $\mathbf{y} = [\mathbf{y}_1, \ldots, \mathbf{y}_M]$ we want to choose the weight matrices $W_k$ for each CW-complex $\mathbf{x}_i$ such that we solve $\min_W \|f(\mathbf{X}) - \mathbf{y}\|_2^2$.

### 3.2 CW-complex Attention

In order to define, a CW-complex Attention Network (CW-AN) we must first develop a notion of connectedness or adjacency for CW-complexes. The analogue of adjacency for CW-complexes is termed incidence. We collect these incidence values, which are integers, in a matrix. This matrix is defined below as $B_k$ in equation (4).

#### 3.2.1 Incidence Matrices

Let the relation $\prec$ denote incidence. If two cells $e_j^{k-1}$ and $e_i^k$ are incident, we write $e_j^{k-1} \prec e_i^k$. Similarly, if two cells $e_j^{k-1}$ and $e_i^k$ are not incident, we write $e_j^{k-1} \not\prec e_i^k$. Additionally, let the relation $\sim$ denote orientation. We write $e_j^{k-1} \sim e_i^k$ if the cells have the same orientation. If two cells have the opposite orientation we

write $e_j^{k-1} \nsim e_i^k$. This enables us to define the classical incidence matrix (Sardellitti & Barbarossa, 2024). Traditionally we define for indices $i$ and $j$ the value of $(B_k)_{j,i}$ as:

$$(B_k)_{j,i} = \begin{cases} 0 & \text{if } e_j^{k-1} \nprec e_i^k \\ 1 & \text{if } e_j^{k-1} \prec e_i^k \text{ and } e_j^{k-1} \sim e_i^k \\ -1 & \text{if } e_j^{k-1} \prec e_i^k \text{ and } e_j^{k-1} \nsim e_i^k \end{cases} \tag{4}$$

The matrix $B_k$ establishes which $k$-cells are incident to which $k-1$-cells. We know from above that $B_k \in \mathbb{Z}^{N_{k-1} \times N_k}$. Let $\mathrm{Col}_j(B_k)$ denote the $j$-th column of $B_k$. We know that $\mathrm{Col}_j(B_k) = \mathbb{Z}^{N_{k-1} \times 1}$ which corresponds to a vector representation of the cell boundary $\partial e_j^k$ viewed as a $k-1$ chain (Alain et al., 2023).

Using the definition of an incidence matrix and the relation $\prec$ we can now define a CW-AN.

**Definition 3.3.** Let $X$ be a finite $n$-dimensional CW-complex. Then for $k \in [0, n]$ we can define a **CW-complex Attention Network (CW-AN)**. A CW-AN computes a learned weighted average of the representations of each skeleton. We start by initializing for all $i$ the starting hidden state $h_{e_i^0}^{(0)} = \left[ e_1^0, \ldots, e_{N_0}^0 \right]^\top \left( \mathrm{Col}_i(B_0^\top) \right)^\top \in \mathbb{R}^{N_0 \times N_0}$. We then define a scoring function $\mathcal{S}$

$$\mathcal{S}(h_{e_i^k}^{(k)}, h_{e_j^{k-1}}^{(k-1)}) = \mathrm{LeakyReLU} \left( \left[ W_k h_{e_i^k}^{(k)} \parallel B_k^\top W_{k-1} h_{e_j^{k-1}}^{(k-1)} B_k \right] a^\top \right) \tag{5}$$

where $W_k \in \mathbb{R}^{N_k \times N_k}$, $W_{k-1} \in \mathbb{R}^{N_{k-1} \times N_{k-1}}$, $B_k \in \mathbb{R}^{N_{k-1} \times N_k}$, $a^\top \in \mathbb{R}^{2N_k \times N_k}$, and $\parallel$ denotes vector concatenation. Checking dimensions, we see that:

$$\dim(W_k h_{e_i^k}^{(k)}) = (N_k \times N_k)(N_k \times N_k) = N_k \times N_k \tag{6}$$

Additionally, for $e_j^{k-1}$ we see:

$$\dim(B_k^\top W_{k-1} h_{e_j^{k-1}}^{(k-1)} B_k) = (N_k \times N_{k-1})(N_{k-1} \times N_{k-1})(N_{k-1} \times N_{k-1})(N_{k-1} \times N_k) = N_k \times N_k \tag{7}$$

By equations (6) and (7) we know:

$$\dim \left( \left[ W_k h_{e_i^k}^{(k)} \parallel B_k^\top W_{k-1} h_{e_j^{k-1}}^{(k-1)} B_k \right] \right) = N_k \times 2N_k \tag{8}$$

Equations (5) and (8) imply:

$$\dim \left( \mathcal{S}(h_{e_i^k}^{(k)}, h_{e_j^{k-1}}^{(k-1)}) \right) = \dim \left( \mathrm{LeakyReLU} \left( \left[ W_k h_{e_i^k}^{(k)} \parallel B_k^\top W_{k-1} h_{e_j^{k-1}}^{(k-1)} B_k \right] a^\top \right) \right) = N_k \times N_k \tag{9}$$

Therefore, we can compute attention scores $\alpha_{e_i^k, e_i^k}$, which are normalized over all incident cells. Let the set $\mathcal{N}_{e_i^k} := \left\{ e_{j'}^{k-1} \mid e_{j'}^{k-1} \prec e_i^k \wedge (e_{j'}^{k-1} \sim e_i^k \vee e_{j'}^{k-1} \nsim e_i^k) \right\}$ contain all cells incident to $e_i^k$. We define a variation of the standard Softmax function called Cell-Softmax, which converts cells to a probability distribution.

$$\alpha_{e_i^k, e_j^{k-1}} = \mathrm{Cell\text{-}Softmax} \left( \mathcal{S}(h_{e_i^k}^{(k)}, h_{e_j^{k-1}}^{(k-1)}) \right) = \frac{\exp \left( \mathcal{S}(h_{e_i^k}^{(k)}, h_{e_j^{k-1}}^{(k-1)}) \right)}{\sum\limits_{e_{j'}^{k-1} \in \mathcal{N}_{e_i^k}} \exp \left( \mathcal{S}(h_{e_i^k}^{(k)}, h_{e_{j'}^{k-1}}^{(k-1)}) \right)} \in \mathbb{R}^{N_k \times N_k} \tag{10}$$

This enables us to define our update rule for computing later hidden states:

$$h_{e_i^k}^{(k)} = \sigma \left( \sum_{e_{j'}^{k-1} \in \mathcal{N}_{e_i^k}} \alpha_{e_i^k, e_{j'}^{k-1}} B_k^\top W_{k-1} h_{e_{j'}^{k-1}}^{(k-1)} B_k \right) \tag{11}$$

We know from equation (7) that

$$\dim(h_{e_i^k}^{(k)}) = \dim\left(\alpha_{e_i^k, e_{j'}^{k-1}} B_k^\top W_{k-1} h_{e_{j'}^{k-1}}^{(k-1)} B_k\right) = (N_k \times N_k)(N_k \times N_k) = N_k \times N_k \qquad (12)$$

Thus, we have defined a notion of self-attention. A CW-AN is then composed by stacking numerous $h_{e_i^k}^{(k)}$ for all $k \in [0, n]$ and all cells.

We note that all $W_i$ and $a^\top$ are learned. To stabilize the learning process of this self-attention mechanism, we can extend definition 3.3 to develop multi-head attention. Just as in Veličković et al. (2018), we can concatenate $K$ independent self-attention mechanisms to execute equation (11). This results in the following representation

$$h_i' = \Big\|_{\ell=1}^{K} \sigma\left(\sum_{e_{j'}^{k-1} \in \mathcal{N}_{e_i^k}} \alpha_{e_i^k, e_{j'}^{k-1}}^{(\ell)} B_k^\top W_{k-1}^{(\ell)} h_{e_{j'}^{k-1}}^{(k-1)} B_k\right) \qquad (13)$$

where $\alpha_{e_i^k, e_{j'}^{k-1}}^{(\ell)}$ are normalized attention coefficients computed by the $\ell$-th attention mechanism and $W_{k-1}^{(\ell)}$ is the corresponding weight matrix. In practice, the operation of the self-attention layer described in definition 3.3 can be parallelized. One can develop a transformer based architecture by combining the multi-head attention layer described in equation (13) with modified Add&Layer Norm as well as Feed Forward networks, equivalent to those developed by Vaswani et al. (2017). The weight matrices $W_k$ can be randomly initialized or one may adopt a different strategy.

## 4 Model Architecture

We develop two distinct networks for one synthetic task.

### 4.1 CW-CNN architecture

Graph convolutional networks are thought of as models which efficiently propagate information on graphs (Kipf & Welling, 2017). Convolutional networks traditionally stack multiple convolutional, pooling, and fully-connected layers to encode image-specific features for image-focused tasks (O'Shea & Nash, 2015). The CW-CNN is motivated by similar principles. In particular, a CW-CNN efficiently propagates information on CW-complexes and encodes topological features for tasks in which geometry plays a pivotal role.

There are numerous potential use cases for such an architecture. In the areas of drug design, molecular modeling, and protein informatics the three dimensional structure of molecules plays a pivotal role and numerous approaches in these areas have attempted to meaningfully extract or utilize geometric information (Kuang et al., 2024; Gebauer et al., 2022; Zhung et al., 2024; Isert et al., 2023).

A CW-CNN is composed of a stack of Convolutional CW-complex layers as described above in definition 3.1. One may additionally add pooling, linear, or dropout layers. In our experiment we stack Convolutional CW-complex layers, and follow up with a Linear layer and GELU activation. The architecture is pictured below.

### 4.2 CW-AT architecture

Initially, Graph Attention Networks were developed as a method to perform node classification on graph-structured data (Veličković et al., 2018). Transformers are neural sequence transduction models that maintain an encoder decoder structure, wherein an input sequence $(x_1, \ldots, x_n)$ is mapped to a sequence of continuous representations $(z_1, \ldots, z_n)$. Then $(z_1, \ldots, z_n)$ are passed to the decoder for generating an output sequence $(y_1, \ldots, y_m)$ in an auto-regressive fashion (Vaswani et al., 2017). The CW-AT is motivated by similar principles. In particular, a CW-AT leverages a different kind of attention mechanism, which can perform classification or regression on CW-complex structured data. While one can theoretically setup a sequence-to-sequence

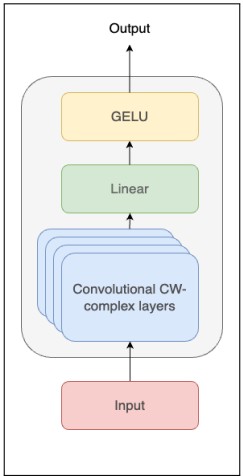

Figure 1: CW-CNN architecture.

learning problem utilizing CW-complexes, we do not venture into such problems. However, we develop an architecture that vaguely resembles the classic Transformer with far fewer parameters.

There are numerous potential use cases for such an architecture. One can attempt language translation, image captioning and sequence transformation tasks (Sutskever et al., 2014). Doing so would require viewing a word or entity as a CW-complex. This is somewhat reasonable since the 1-skeleton of a CW-complex is a graph without loops (Whitehead, 1949). Graphs have appeared in numerous contexts within Natural language processing. Classically vertices can encode text units of various sizes and characteristics such as words, collocations, word senses, sentences and documents (Nastase et al., 2015). Edges may represent relationships such as co-occurrence (Nastase et al., 2015). One can replace the notion of vertex with cell and edge with a kind of gluing map and extend these ideas to CW-complexes. One can represent co-occurence for instance by scaling the matrix $B_k$.

A CW-AT is composed of a single network, receiving as input a CW-complex. The input is processed by a Multi-Cellular Attention mechanism as described by equation (13). Afterwards one may apply dropout, and layer norm. Finally, a linear layer followed by Add and SELU are used to get the desired output shape. One may apply a Softmax if the output is to be viewed as probabilities.

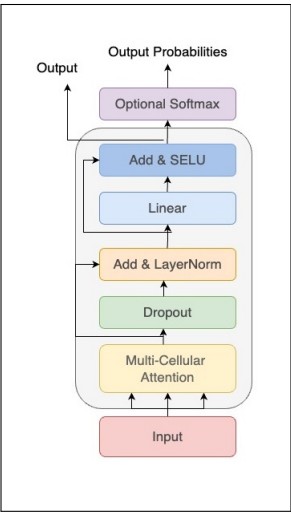

Figure 2: CW-AT architecture.

## 5 Experiments

In this section we conduct experiments solving real-world graph classification problems. We focus on the frequently-used benchmarks from TUDataset (Morris et al., 2020). We include the following datasets: Proteins, MUTAG, PTC, NCI1, and NCI109. The Proteins dataset is from biology. The MUTAG, PTC, NCI1 are from chemistry. The datasets store all molecules as graphs. Datasets in which the input is represented as a CW-complex are not widely available. In order to run these experiments we must pre-process the graphs to be compatible with our architecture, however doing so is not ideal. This is due to the fact that there is no natural way to represent 3-dimensional structures of molecules and many other chemical properties essential to the molecule's functionality simply through molecular graphs (Liu et al., 2022).

Table 2: Experimental Results

| Dataset: | MUTAG | PTC | Proteins | NCI1 | NCI109 |
|---|---|---|---|---|---|
| CWN (Bodnar et al., 2021) | ? | ? | ? | ? | ? |
| CXN (Hajij et al., 2021) | ? | ? | ? | ? | ? |
| CAN (Giusti et al., 2023) | ? | ? | ? | ? | ? |
| **CW-AT** | ? | ? | ? | ? | ? |
| **CW-CNN** | ? | ? | ? | ? | ? |

From the above table we can see.

## 6 Conclusion

In this work, we presented the CW-CNN and CW-AT, the first types of neural network that can receive CW-complexes as input. We demonstrate reasonable accuracy with relatively few parameters on real-world tasks. These results have implications for machine learning tasks in which geometric information or three dimensional structure plays a pivotal role. These areas include, but are not limited to, molecular design, cheminformatics and drug discovery. Additionally one may view tasks involving graphs in natural language as good candidates for a CW-complex representation and correspondingly a CW-AT. CW-complexes capture interactions between higher-order cells enabling the one to model polyadic relations effectively. Our neural networks enable learning on CW-complexes thereby facilitating the learning of polyadic relations. We are excited about the future of these models and plan to apply them to other tasks in cheminformatics and natural language processing.

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

# A Appendix

## A.1 Mathematical Definitions for CW-Complexes

We provide the definition given by Alain et al. (2023) and re-use some of their notation. In order to define a path and function over a finite cellular complex or CW-complex $X$, one has to define a notion of chains and cochains.

**Definition A.1.** Suppose $X$ is an $n$-dimensional complex. Then, a $k$-chain $c_k$ for $0 \leq k \leq n$ is simply a sum over the cells: $c_k = \sum_{i=1}^{N_k} \eta_i e_i^k$, $\eta_i \in \mathbb{Z}$. The authors show this generalizes the notion of directed paths on a graph. The set of all $k$-chains on $X$ is denoted by $C_k(X)$, which has the algebraic structure of a free Abelian group with basis $\{e_i^k\}_{i=1}^{N_k}$.

**Definition A.2.** Given definition A.1, the boundary operator naturally follows as $\partial_k : C_k(X) \to C_{k-1}(X)$. The operator $\partial_k$ maps the boundary of a $k$-chain to a $k-1$-chain. This map is linear, thereby leading to the equation: $\partial_k \left( \sum_{i=1}^{N_k} \eta_i e_i^k \right) = \sum_{i=1}^{N_k} \eta_i \partial(e_i^k)$.

The authors then define the $k$-cochain (the dual notion of the $k$-chain) and the coboundary operator (the dual notion of the boundary operator).

**Definition A.3.** Suppose $X$ is an $n$-dimensional complex. Then, a $k$-cochain on $X$ is a linear map $f : C_k(X) \to \mathbb{R}$ where $0 \leq k \leq n$. $f \left( \sum_{i=1}^{N_k} \eta_i e_i^k \right) = \sum_{i=1}^{N_k} \eta_i f(e_i^k)$ where $f(e_i^k) \in \mathbb{R}$ is the value of $f$ at cell $e_i^k$. The space of $k$-cochains is defined as $C^k(X)$, which forms a real vector space with the dual basis $\{(e_i^k)^*\}_{i=1}^{N_k}$ such that $(e_i^k)^*(e_j^k) = \delta_{ij}$.

**Definition A.4.** Given definition A.3, the coboundary operator naturally follows as $d_k : C^k(X) \to C^{k+1}(X)$ which, for $0 \le k \le n$, is defined as $d_k = f(\partial_{k+1}(c))$ for all $f \in C^k(X)$ and $c \in C_{k+1}(X)$. Note that for $k \in \{-1, n\}$ $d_k f \equiv 0$.

Using these definitions, Alain et al. (2023) formally introduce a generalization of the Laplacian for graphs. They further prove that for $k = 0$ and identity weights, the Hodge Laplacian is the graph Laplacian. In essence, the authors prove $W_0 = I \implies \Delta_0 = B_1 W_1 B_1^\top$ and $W_1 = I \implies \Delta_0 = B_1 B_1^\top$, which is the standard graph Laplacian.

**Definition A.5.** Let $X$ be a finite complex. Then, we define a set of weights for every $k$. Namely, let $\{w_i^k\}_{i=1}^{N_k}$ be a set of real valued weights. Then, $\forall f, g \in C^k(X)$, one can write the weighted $L^2$ inner product as: $\langle f, g \rangle_{L^2(w^k)} := \sum_{i=1}^{N_k} w_i^k f(e_i^k) g(e_i^k)$. This inner product induces an adjoint of the coboundary operator $d_k^* : C^{k+1}(X) \to C^k(X)$. Namely, $\langle d_k^* f, g \rangle = \langle f, d_k g \rangle$ for all $f \in C^{k+1}(X)$ and $g \in C^k(X)$.

**Definition A.6.** Using the previous definitions, the Hodge Laplacian $\Delta_k : C^k(X) \to C^k(X)$ on the space of $k$-cochains is then $\Delta_k := d_{k-1} \circ d_{k-1}^* + d_k^* \circ d_k$. The matrix representation is then $\Delta_k := B_k^\top W_{k-1}^{-1} B_k W_k + W_k^{-1} B_{k+1} W_{k+1} B_{k+1}^\top$. Here, $W_k = \mathrm{diag}(w_1^k, \ldots, w_{N_k}^k)$ is the diagonal matrix of cell weights and $B_k$ is the order $k$ incidence matrix, whose $j$-th column corresponds to a vector representation of the cell boundary $\partial e_j^k$ viewed as a $k-1$ chain.

