# OpenReview forum: "CW-CNN & CW-AN: Convolutional Networks and Attention Networks for CW-Complexes"
_TMLR — Rejected by TMLR_

### Review · Reviewer_HeyR · 2024-09-03

**Summary Of Contributions:**

The authors present representation learning frameworks for CW-complexes with graph convolution and attention mechanism.

**Audience:**

No

**Broader Impact Concerns:**

Please see Weakness.

**Claims And Evidence:**

No

**Requested Changes:**

Please see Weakness.

**Strengths And Weaknesses:**

Weakness:

Extending GCN and GAT to CW-complexes is not new. For example, “Mustafa Hajij, Kyle Istvan, and Ghada Zamzmi. Cell complex neural networks. NeurIPS 2020 Workshop TDA and Beyond, 2020.”, “Cristian Bodnar, Fabrizio Frasca, Nina Otter, Yu Guang Wang, Pietro Liò, Guido F Montufar, and Michael Bronstein. Weisfeiler and lehman go cellular: Cw networks. Advances in Neural Information Processing Systems, 34, 2021” define message passing on CW-complexes. “L. Giusti, C. Battiloro, L. Testa, P. Di Lorenzo, S. Sardellitti and S. Barbarossa, "Cell Attention Networks," 2023 International Joint Conference on Neural Networks (IJCNN), “ defines the attention mechanism on cell complexes.

The concept of utilizing incidence matrix and Hodge laplacians to define neural networks also widely exists in the literature. What is the difference between the methods proposed in this submission and the related works? I don’t see a difference with the previous works. If this approach is indeed new, this submission needs a discussion of related works that are currently missing. This is the major issue that needs to be clarified and addressed. Besides, there are many papers on representation learning for simplicial complexes, which are also relevant to this topic.

The experiment section is fragile. The author claims that there’s no baseline but many related works as mentioned above exist. Also, the synthetic dataset needs more description such as the generating process and the build of features.

---

> ### Author Response · Authors · 2024-09-03
> **Re: Reviewer HeyR**
>
> We thank the reviewer for their insightful comments.
>
> 1.  “Mustafa Hajij, et al. Cell complex neural networks. NeurIPS 2020 Workshop, 2020.
>
> We highlight key differences with this work in the first paragraph of our updated manuscript (2.4).
>
> 2. “Cristian Bodnar, et al. Weisfeiler and lehman go cellular: CW networks. NeurIPS, 34, 2021
>
> We highlight key differences with this work in the second paragraph of our updated manuscript (2.4).
>
> 3. “L. Giusti, C. Battiloro, et al, "Cell Attention Networks," 2023 International Joint Conference on Neural Networks (IJCNN)"
>
> We highlight key differences with this work in the third paragraph of our updated manuscript (2.4).
>
> 4. The concept of utilizing Hodge-laplacians to define neural networks also widely exists in the literature.
>
> Please provide a source for CW-complexes. We do not believe one exists.
>
> 5. Literature review
>
> We agree that the literature review should be improved. We do so in the updated manuscript section 2.4.
>
> 6. Experimentation
>
> This is discussed in more depth in the comment below.

---

> ### Author Response · Authors · 2024-09-03
> **Re: Reviewer HeyR (Updated Manuscript)**
>
> We thank reviewer HeyR for their feedback and suggestions.
>
> We update our manuscript to address the following questions proposed by Reviewer HeyR:
> 1. What is the difference between the methods proposed in this submission and the related works?
>
> We include a discussion of all mentioned related works and make explicit the distinctions between our work and that of others. We briefly highlight the theoretical advantages of our approach as well. These changes are primarily concentrated in section 2.4 and are typeset in blue. This is a refinement of our prior comment.
>
> Hopefully we have addressed a major issue that needs to be clarified.
>
> 2. Experiments:
>
> CCN[1]: Mustafa Hajij, Kyle Istvan, and Ghada Zamzmi. Cell complex neural networks,
>
> WLG[2]: Cristian Bodnar, Fabrizio Frasca, Yu Guang Wang, Nina Otter, Guido Montúfar, Pietro Liò, and Michael Bronstein. Weisfeiler and Lehman Go Cellular: CW Networks
>
> CAN[3]: Lorenzo Giusti and Claudio Battiloro and Lucia Testa and Paolo Di Lorenzo and Stefania Sardellitti and Sergio Barbarossa, Cell Attention Networks
>
> LIU[4]: Lu S., Wang, H., Liu, W., Lasenby, J., Guo, H., and Tang, J. Pre-training molecular graph representation with 3d geometry. In International Conference on Learning Representations (2022).
>
> In the context of experiments CCN[1] does not provide any experimental results. WLG[2] and CAN[3] utilize lifting maps on graphs. In the experiments section of both papers the data is encoded as graphs or converted to graphs (MUTAG, PTC, Proteins, NCI1, IMDB-B etc.). Datasets in which the input is represented as a CW-complex are not widely available. One can preprocess the graphs to be compatible with our architecture, however doing so is not ideal. Take for example the MUTAG dataset. MUTAG is a collection of nitroaromatic compounds and the goal is to predict their mutagenicity on Salmonella typhimurium. One can represent a nitroaromatic compound as a molecular graph. However, doing so poses issues. As described by Liu et al (2022), there is no natural way to represent 3-dimensional structures of molecules and many other chemical properties essential to the molecule’s functionality simply through molecular graphs [4]. This, in fact, has prompted the development of a different kind of representation for molecules. Hence, explaining why we find doing so to be "not ideal."

---

> > ### Comment · Reviewer_HeyR · 2024-10-17
> > **Comment**
> >
> > I thank the reviewer for the response and revision. I hold similar opinion with the other reviewers on comparison with existing works, real-world datasets and add baselines.

---

> > > ### Author Response · Authors · 2024-10-21
> > > **re: Comment**
> > >
> > > (1) This has been addressed in detail. Please see the revised manuscript
> > >
> > > (2) This is currently underway -> the training job is running
> > >
> > > (3) Due to recent model simplifications, I don't think there is much value in an ablation study. Effectively removing a linear layer, and dropout shouldn't alter very much. Removing the attention or convolution mechanism altogether seems pointless.

---

### Review · Reviewer_3rbq · 2024-09-27

**Summary Of Contributions:**

This paper presents a novel framework for learning representations on data with CW-complex structures. In particular, they discuss the notions of convolution and attention for CW-complexes, and them use them to design HodgeLaplacian informed neural network for CW-structured data.

**Audience:**

Yes

**Claims And Evidence:**

No

**Requested Changes:**

**Recommendation:** Including a dataset with 2 or 3 CW complex structures and conducting a comparison with existing GNNs (using only 1D CW complex as graph structures) would strengthen the paper's significance and provide a more convincing evaluation of the proposed method.

**Strengths And Weaknesses:**

## Strengths:

- The paper's purpose is well-defined and easy to follow.
- The proposed extensions to attention and convolution for CW-structured data are original and presented in a clear and concise manner.

## Weaknesses:

- While the extensions are important, their implementation is relatively straightforward, potentially limiting the overall contribution.
- The paper could benefit from a more comprehensive comparison to existing graph neural networks, especially on datasets with higher-dimensional CW complex structures. This would help demonstrate the advantages of learning with high-dimensional cells.

---

> ### Author Response · Authors · 2024-10-05
> **Re: Reviewer 3rbq**
>
> We thank Reviewer 3rbq for their insightful comments.
>
> 1. The paper could benefit from a more comprehensive comparison to existing graph neural networks, especially on datasets with higher-dimensional CW complex structures. This would help demonstrate the advantages of learning with high-dimensional cells.
>
> We agree with the reviewer that a more comprehensive comparison is in order. We additionally appreciate the reviewer's suggestion and would like to conduct additional experiments. However, we one question.
>
> Reviewer Suggestion: Including a dataset with 2 or 3 CW complex structures and conducting a comparison with existing GNNs (using only 1D CW complex as graph structures) would strengthen the paper's significance and provide a more convincing evaluation of the proposed method.
>
> To clarify, Is the reviewer suggesting that we build or find a dataset consisting of 2 or 3 dimensional CW-complexes. Then, feed in the one skeleton's of all datapoints to a GNN and feed in the entire structure to our model and report the accuracy?
>
> If this is the case we can implement such an experiment and add a table/experimental setup.

---

> > ### Comment · Reviewer_fyM9 · 2024-10-10
> > **adding more comparision**
> >
> > Maybe the authors can add a thorough experimentations, more theoretical analysis on the comparison between this model and existing models. There is also a similarity between this work and weighted cell complexes. You should also add.

---

> > > ### Author Response · Authors · 2024-10-11
> > > **re: more comparison**
> > >
> > > We thank the reviewer for their feedback.
> > >
> > > These points are actionable and I agree with the suggestions. I will repeat what I told another reviewer.
> > >
> > > (1) I will add a few more paragraphs not only describing the differences between my work theoretically but also intuitively. I will add table(s) in addition to compare the intuitive and theoretical aspects.
> > >
> > > (2) I agree that there should be more dataset comparisons. I will run my models on some adapted cell complex datasets / some used in other papers. Currently I'm thinking of the ones used in the Bodnar paper.
> > >
> > > (3) I can run an ablation study no problem.
> > >
> > > (4) weighted cell complexes: The only context in which I can find "weighted" and "cw complexes" is actually a paper on weighted simplical complexes by Baccini, Geraci, and Bianconi or in the context of weighted discrete morse theory which seems a bit irrelevant.  If there's something I'm missing or a specific paper you had in mind I can discuss that in the literature review. I can also add the weighted simplical complexes work to the literature review.

---

> > > > ### Comment · Reviewer_fyM9 · 2024-10-11
> > > > **compare against the three existing paper**
> > > >
> > > > I think you need to compare against three existing papers, Bodnar, Hajij and Battiloro (Cell Attention Networks). The ones that do no have existing implementation online can be implemented easily. I think Topox has implementations for these models.

---

> > > > > ### Author Response · Authors · 2024-10-21
> > > > > **re: comparison against three paper**
> > > > >
> > > > > (1) This has been addressed in detail. Please see the revised manuscript
> > > > >
> > > > > (2) This is currently underway -> the training job is running
> > > > >
> > > > > (3) Due to recent model simplifications, I don't think there is much value in an ablation study. Effectively removing a linear layer, and dropout shouldn't alter very much. Removing the attention or convolution mechanism altogether seems pointless.

---

### Review · Reviewer_fyM9 · 2024-10-08

**Summary Of Contributions:**

The manuscript addresses the learning problem involving a dataset \( D = \{(x_i, y_i)\}_{i=1}^n \), where \( x_i \) is a CW-complex and \( y_i \in \mathbb{R}^d \). The authors aim to learn a function \( F \) such that \( y_i = F(x_i) + \epsilon \). They propose convolutional and attention mechanisms specifically designed for CW-complexes, building on the foundational works of Kipf & Welling (2017) and Veličković et al. (2018). However, the proposed networks appear to be trivial special cases of existing approaches, particularly resembling the work of Hajij and Bodnar.

**Audience:**

No

**Broader Impact Concerns:**

Identical to previous work, no impact.

**Claims And Evidence:**

No

**Requested Changes:**

The authors seem to have addressed he previous work by another reviewer. However, almost no work has been done to make the paper actually original. It is a previous work.

**Strengths And Weaknesses:**

1. **Lack of Novelty:** The proposed convolutional and attention mechanisms do not offer significant advancements beyond what is already established in the literature. The adaptations made from the works of Kipf & Welling (2017) and Veličković et al. (2018) do not clearly progress the state of the art and seem to be trivial special cases of existing methods.

2. **Similarity to Existing Work:** The proposed networks closely resemble the methodologies presented by Hajij and Bodnar, which raises concerns about the originality and contributions of this manuscript. The lack of differentiation from existing approaches undermines the novelty of the proposed solutions.

3. **Insufficient Comparisons:** The manuscript does not provide comprehensive comparisons with existing models in the domain of cell complex neural networks. This omission makes it challenging to assess the effectiveness and relevance of the proposed methods, as comparisons are essential for evaluating their impact.

---

> ### Author Response · Authors · 2024-10-08
> **Re: Reviewer fyM9**
>
> We thank the reviewer for their insightful comments.
>
> 1. Lack of Novelty: The proposed convolutional and attention mechanisms do not offer significant advancements beyond what is already established in the literature. The adaptations made from the works of Kipf & Welling (2017) and Veličković et al. (2018) do not clearly progress the state of the art and seem to be trivial special cases of existing methods.
>
> The convolutional and attention mechanisms are in fact different. Suggesting that they are trivial special cases blatantly contradicts the definitions outlined in sections 2.1, 2.2 and 2.4. If they are trivial special cases, can you start with the provided equations from Hajij and Bodnar and derive my equation?
>
> 2. Similarity to Existing Work: The proposed networks closely resemble the methodologies presented by Hajij and Bodnar, which raises concerns about the originality and contributions of this manuscript. The lack of differentiation from existing approaches undermines the novelty of the proposed solutions.
>
> None of the existing methods are Hodge-aware. Hajij proposes training n-1 many networks for for each of the k-cells independently. We propose training only 1 network which results in a linear improvement in overall architecture size. We also don't rely on permutation equivariant/differentiable functions in the same way as Hajij instead substituting that role with the boundary operators. Bodnar defines lifting transformations to perform attention and uses this as the basis for their message passing scheme. We do not rely on lifting transformations to map from a graph to regular cell complex. As a result, our scheme is mathematically different. This was outlined in depth in section 2.4.
>
> 3. Insufficient Comparisons
>
> This I somewhat agree with. Do you have a suggested set of comparisons or experiments that would make the results more convincing?
>
> Requested changes: "However, almost no work has been done to make the paper actually original. It is a previous work. / Identical to previous work"
>
> The presented statement is objectively false. The formulae presented are different.

---

> > ### Comment · Reviewer_fyM9 · 2024-10-08
> > **message passing**
> >
> > the statement that I provided about your model being a special case is mathematically correct. Your model is a special case of the message passing paradigm which is suggested by Hajij and Bodnar (both for boundary/coboundary (co)adj ) which includes the Hodge as a special case. You have a one line comparison to other existing line of work and you only acknowledge prior work after being told to do so. I think this paper is not on the level publishing in this venue.

---

> ### Author Response · Authors · 2024-10-08
> **Re: message passing**
>
> I agree it is a special case of message passing. You could argue Bodnar's paper (Jun, 2021) is a special case of the message passing paradigm that Hajij first introduced in October of 2020. That doesn't make Bodnar's work "Identical" to Hajij's work or not novel. My formula is not a trivial version of theirs, and our mechanisms are different (both convolution and attention). They do not explicitly introduce boundary/coboundary and the Hodge laplacian in the context of CW-complexes. I respect your opinion however.
>
> We could say that every GNN paper that has ever been published is just a special case of the message passing paradigm. To be quite honest, the reason why "I only acknowledge prior work after being told to do so" is because I had no idea the prior work even existed until after it was pointed out/post submission. My literature review initially wasn't up to the standard. I fixed that by addressing the first reviewers points. Even then my work is different. I read those papers multiple times to make sure of that. If you think my evaluation of other architectures is limited, I am happy to compare.

---

> > ### Comment · Reviewer_fyM9 · 2024-10-11
> > **More experimental work is needed**
> >
> > I see you point and they are fair.
> >
> > I think the only thing needed is the following
> >
> > (1) explain very clearly the difference between your work and exiting work. Intuitively and theoretically.
> >
> > (2) run a fair analysis on cell complex datasets comparing existing models and yours. Report the results.
> >
> > (3) run ablation studies.

---

> > > ### Author Response · Authors · 2024-10-11
> > > **Re: more experimental work**
> > >
> > > We thank the reviewer for their feedback.
> > >
> > > These points are actionable and I agree with the suggestions. Here is my proposed re-working:
> > >
> > > (1) I will add a few more paragraphs not only describing the differences between my work theoretically but also intuitively. I will add table(s) in addition to compare the intuitive and theoretical aspects.
> > >
> > > (2)  I agree that there should be more dataset comparisons. I will run my models on some adapted cell complex datasets / some used in other papers. Currently I'm thinking of the ones used in the Bodnar paper.
> > >
> > > (3) I can run an ablation study no problem.

---

> ### Author Response · Authors · 2024-10-21
> **re: more experimental work (2)**
>
> (1) This has been addressed in detail. Please see the revised manuscript
>
> (2) This is currently underway -> the training job is running
>
> (3) Due to recent model simplifications, I don't think there is much value in an ablation study. Effectively removing a linear layer, and dropout shouldn't alter very much. Removing the attention or convolution mechanism altogether seems pointless.

---

> > ### Comment · Reviewer_fyM9 · 2024-11-01
> > **some corrections**
> >
> > I think the authors addressed most of the issues. There are some still left.
> >
> > (1) The authors did not review the work by CWN, CXN and CAN well. For instance, the authors did not explain the message passings given by the first two papers (both give equivalent message passing, towards the end), nor the generalitty of the second method.
> > (2) Both CWN and CXN  are message passing cellular networks that can be parameterized in the special case to MLP or conv layers but they do not have to. So please add the message passing as well.
> > (3) You should add all implementations and compare against all methods. you can find them here : https://github.com/pyt-team/TopoModelX/tree/main/topomodelx/nn/cell

---

> > > ### Author Response · Authors · 2024-11-03
> > > **re: some corrections - addressed (1) and (2) see revised manuscript**
> > >
> > > We have addressed (1) and (2). We explained the full message passing algorithms for all three papers. We discussed the generality of the second method. We discussed the possible parameterizations of all three methods. All these distinctions are given as separate paragraphs for each model in section 2.4.2. Please see the updated manuscript. We are having some slight technical issues with the package the reviewers suggested (Topox). We are working on resolving them.

---

> ### Author Response · Authors · 2024-11-01
> **re: some corrections**
>
> We thank the reviewer for their feedback.
>
> We agree with the suggested corrections and will make the suggested changes (1)-(3). (1) and (2) will be added to 2.5.2 and (3) will be added using Topox.

---

### Comment · Reviewer_fyM9 · 2024-10-08
**seem pretty similar to existing networks on Cell complex**

Review Summary for Manuscript: "CW-CNN & CW-AN: Convolutional Networks and Attention Networks for CW-Complexes":

Summary:

The manuscript addresses the learning problem involving a dataset \( D = \{(x_i, y_i)\}_{i=1}^n \), where \( x_i \) is a CW-complex and \( y_i \in \mathbb{R}^d \). The authors aim to learn a function \( F \) such that \( y_i = F(x_i) + \epsilon \). They propose convolutional and attention mechanisms specifically designed for CW-complexes, building on the foundational works of Kipf & Welling (2017) and Veličković et al. (2018). However, the proposed networks appear to be trivial special cases of existing approaches, particularly resembling the work of Hajij and Bodnar.

Recommendation :
I recommend rejecting this manuscript for the following reasons:

1. **Lack of Novelty:** The proposed convolutional and attention mechanisms do not offer significant advancements beyond what is already established in the literature. The adaptations made from the works of Kipf & Welling (2017) and Veličković et al. (2018) do not clearly progress the state of the art and seem to be trivial special cases of existing methods.

2. **Similarity to Existing Work:** The proposed networks closely resemble (almost identical) the methodologies presented by Hajij and Bodnar (and others suggested similar work on simplicial complexes and other complexes), which raises concerns about the originality and contributions of this manuscript. The lack of differentiation from existing approaches undermines the novelty of the proposed solutions.

3. **Insufficient Comparisons:** The manuscript does not provide comprehensive comparisons with existing models in the domain of cell complex neural networks. This omission makes it challenging to assess the effectiveness and relevance of the proposed methods, as comparisons are essential for evaluating their impact.

Given these points, the paper does not meet the standards expected for publication in TMLR. I encourage the authors to address these concerns and consider resubmitting to a more appropriate venue that aligns with the current level of innovation presented in their work.

---

> ### Author Response · Authors · 2024-10-08
> **same as above**
>
> We thank the reviewer for repeating their insightful comments.
>
> Lack of Novelty: The proposed convolutional and attention mechanisms do not offer significant advancements beyond what is already established in the literature. The adaptations made from the works of Kipf & Welling (2017) and Veličković et al. (2018) do not clearly progress the state of the art and seem to be trivial special cases of existing methods.
> The convolutional and attention mechanisms are in fact different. Suggesting that they are trivial special cases blatantly contradicts the definitions outlined in sections 2.1, 2.2 and 2.4. If they are trivial special cases, can you start with the provided equations from Hajij and Bodnar and derive my equation? Oh wait you can't because I don't use lifting maps like Bodnar and multiplying by the boundary matrices + applying any nonlinearity isn't necessarily permutation invariant which is a condition of Hajij's equations.
>
> Similarity to Existing Work: The proposed networks closely resemble the methodologies presented by Hajij and Bodnar, which raises concerns about the originality and contributions of this manuscript. The lack of differentiation from existing approaches undermines the novelty of the proposed solutions.
> None of the existing methods are Hodge-aware. Hajij proposes training n-1 many networks for for each of the k-cells independently. We propose training only 1 network which results in a linear improvement in overall architecture size. We also don't rely on permutation equivariant/differentiable functions in the same way as Hajij instead substituting that role with the boundary operators. Bodnar defines lifting transformations to perform attention and uses this as the basis for their message passing scheme. We do not rely on lifting transformations to map from a graph to regular cell complex. As a result, our scheme is mathematically different. This was outlined in depth in section 2.4.
>
> Insufficient Comparisons
> This I somewhat agree with. Do you have a suggested set of comparisons or experiments that would make the results more convincing?

---

### Comment · Reviewer_fyM9 · 2024-11-12
**a few more comments**

First I want to congratulate the authors for doing great job on improving the paper. Indeed now the paper is much better, and it explains well the contribution.

I want to add a few more remarks just to make the lit review more accurate. Also the experiments are not yet in the table ?

(1) In table 1, CXN and CWN are message passing as well as what you have, please add that.
(2) CXN has two more message passing equations, please add them.
(3) CAN message passing is obtained from [1] below. Please read the CAN paper carefully to make accurate the citation at the right place.
(4) Table 2, are you gonna make the final experiment available ?

[1] Hajij, Mustafa, Ghada Zamzmi, Theodore Papamarkou, Nina Miolane, Aldo Guzmán-Sáenz, Karthikeyan Natesan Ramamurthy, Tolga Birdal et al. "Topological deep learning: Going beyond graph data." arXiv preprint arXiv:2206.00606 (2022).

---

### Decision · Action_Editor_6svX · 2024-12-07

**Recommendation:** Reject

**Comment:**

This paper studies models built to handle data that is structured according to CW complexes, a class of topological objects. The goal is to build neural networks that can address these forms of data, which requires generalizing notions from other models (in particular graph neural networks) in multiple ways. For example, the goal is to generalize operations like convolution to work with these forms of data.

The authors present the start of some new variations of performing these operations and discuss some similarities and differences.

All reviewers agree that the paper is not quite ready. However, the initial discussion during the review phase was useful. For example, the reviewers brought up a number of very closely related works, which are now discussed in the draft. This is a critical step.

Several key pieces remain to be done:
- One of the main innovations of the work is to ensure that models and constituent operations are Hodge-aware, but we need some more motivation for why this is useful. This is doable, but will require lifting some of the arguments from Yang & Isufi.
- The experimental evidence is not yet complete; as mentioned by the reviewers, this is the most important step.
- Finally, a minor point: the paper requires a few writing passes.

**Audience:**

Yes, this paper tackles topics of interest to the machine learning community.

**Claims And Evidence:**

No, some of the necessary evidence is not yet present (see below).

**Resubmission Of Major Revision:**

The authors may consider submitting a major revision at a later time.

---

> ### Author Response · Authors · 2024-12-07
> **Re: Decision**
>
> We want to thank the editor and reviewers for their comprehensive feedback!
>
> We will incorporate revised experiments and all the feedback the reviewers provided.
>
> We will re-work the changes into a revised draft and resubmit a major revision later.
>
> Thank you again!